# Assessing the operational feasibility and acceptability of an inhalable formulation of oxytocin for improving community-based prevention of postpartum haemorrhage in Myanmar: a qualitative inquiry

Kyu Kyu Than,[1,2] Victoria Oliver,[3] Yasmin Mohamed,[1,4] Thazin La,[1] Pete Lambert,[3] Michelle McIntosh,[3] Stanley Luchters[1,4,5]

For numbered affiliations see end of article.

**Correspondence to**
Professor Stanley Luchters;
stanley.luchters@burnet.edu.au

## ABSTRACT

**Objective** This study assessed the potential operational feasibility and acceptability of a heat-stable, inhaled oxytocin (IOT) product for community-based prevention of postpartum haemorrhage in Myanmar.

**Methods** A qualitative inquiry was conducted between June 2015 and February 2016 through focus group discussions and in-depth interviews. Research was conducted in South Dagon township (urban setting) and in Ngape and Thanlyin townships (rural settings) in Myanmar. Eleven focus group discussions and 16 in-depth interviews were conducted with mothers, healthcare providers and other key informants. All audio recordings were transcribed verbatim in Myanmar language and were translated into English. Thematic content analysis was done using NVivo software.

**Results** Future introduction of an IOT product for community-based services was found to be acceptable among mothers and healthcare providers and would be feasible for use by lower cadres of healthcare providers, even in remote settings. Responses from healthcare providers and community members highlighted that midwives and volunteer auxiliary midwives would be key advocates for promoting community acceptance of the product. Healthcare providers perceived the ease of use and lack of dependence on cold storage as the main enablers for IOT compared with the current gold standard oxytocin injection. A single-use disposable device with clear pictorial instructions and a price that would be affordable by the poorest communities was suggested. Appropriate training was also said to be essential for the future induction of the product into community settings.

**Conclusion** In Myanmar, where home births are common, access to cold storage and skilled personnel who are able to deliver injectable oxytocin is limited. Among community members and healthcare providers, IOT was perceived to be an acceptable and feasible intervention for use by lower cadres of healthcare workers, and thus may be an alternative solution for the prevention of postpartum haemorrhage in community-based settings in the future.

### Strengths and limitations of this study

► This is the first qualitative inquiry exploring the operational feasibility and acceptability of a future heat-stable, inhaled oxytocin (IOT) product for community-based prevention of postpartum haemorrhage in Myanmar to inform product development.

► The participants involved healthcare workers and stakeholders from various levels of the healthcare system and the community and from both urban and rural areas to provide an insight into a range of different perspectives, which has been triangulated to enrich the findings.

► Since the study was conducted in selected townships of Myanmar, the views and opinions towards IOT product may differ in other places of the country.

► Operational feasibility and acceptability of the IOT product was assessed with a similar drug delivery device but in the absence of the final product and in the context of limited experience with inhalable medications, which could have attributed to overwhelmingly positive perspectives.

### BACKGROUND

Postpartum haemorrhage (PPH) is the leading cause of maternal mortality worldwide, contributing to approximately 20% of total maternal deaths.[1] The gold standard uterotonic agent, oxytocin, has been shown to reduce the risk of PPH by 50% and is recommended by the WHO for administration via intravenous or intramuscular injection to every woman during the third stage of labour for the prevention of PPH.[2,3]

Several factors currently limit widespread access to oxytocin in developing countries where the majority of PPH deaths occur.[4] In the injectable form, oxytocin is sensitive

to heat and must be supplied and stored under cool or cold conditions in order to maintain drug potency.[5 6] This presents a significant problem particularly in rural areas where the availability of cold chain storage is limited. The necessity to administer oxytocin via an injection further limits widespread access to this intervention. In low-income and middle-income countries, a significant proportion of births occur outside healthcare facilities, in the absence of the equipment and medical personnel required to deliver an injection.[7–9]

An alternative to injectable oxytocin is misoprostol. This is a heat-stable oral drug, which has been recommended by WHO in settings where injectable oxytocin is not available. However, it is less effective with more frequent side effects compared with oxytocin, and thus the WHO recommends that availability of misoprostol should not detract from efforts to make oxytocin universally accessible.[2 10] In addition, uptake of misoprostol has faced significant challenges in some territories due to policymakers' concerns around abortion misuse and reluctance for task-shifting of uterotonics to community-based healthcare providers who have been the primary target for many misoprostol implementation programmes.[11 12]

Therefore, Monash University and GlaxoSmithKline are currently collaborating to develop a heat-stable, easy-to-use, low-cost, dry powder oxytocin product for respiratory delivery to prevent PPH in resource-constrained settings.[13 14] The product is being designed to simplify the administration of oxytocin and remove the need for refrigerated storage or additional consumables for administration. Inhaled oxytocin (IOT) therefore has the potential to extend on the value proposition of misoprostol by providing the gold standard uterotonic in a format that is suitable for use in settings where access to a reliable cold chain or skilled birth attendant is limited. The development of the IOT product is currently at an early clinical trials stage with pharmacokinetic data in non-pregnant female volunteers indicating that comparable systemic exposure of oxytocin with that achieved following a 10 IU intramuscular injection can be delivered through inhaled delivery.[14] While these data indicate that an inhaled product has the potential to be similarly effective to the current injection, further clinical trials are required and will be conducted to characterise the product fully and determine utility in low-resource settings. However, there are many examples of promising global health technologies that have failed to achieve significant uptake and impact, despite possessing the technical attributes required to address an unmet need.[15 16] Thus, in addition to clinical efficacy, consideration must be given to the end-user acceptability and feasibility of product use, as these will be equally important drivers of the ability of IOT to deliver impact in target territories.

Myanmar represents a territory where the IOT product has the potential to improve maternal health outcomes. The country has one of the highest maternal mortality ratios in the Southeast Asian region at 282 maternal deaths per 100 000 live births.[17] The main cause of maternal mortality is PPH, which is estimated to account for 30% of maternal deaths.[18] Approximately 70% of the population resides in rural areas and the majority of the childbirths occur at home.[19] The health system in Myanmar is divided into central-level, state/division-level and township-level services. At the township level, healthcare services are provided through a township hospital, station hospital, rural health centres and subrural health centres. A township medical officer is responsible for administrative and clinical decision making. Maternal and child health services at the subcentre and community-level are provided by a midwife (MW). Due to shortages in MWs, a volunteer cadre of auxiliary midwives (AMWs) at the community-level support the MWs with provision of maternal and child health services, including care for uncomplicated deliveries.

Approximately one-third of all deliveries in Myanmar are attended out-of-facility settings by an MW or AMW.[20] However, these providers have limited capacity to actively manage the third stage of labour.[21 22] Government policy prohibits MWs and AMWs from delivering injections outside of the health facility setting and thus currently MWs (but not AMWs) are authorised to use misoprostol for routine prevention of PPH and can only deliver an injection of oxytocin for 'life-saving' purposes. AMWs face further restrictions and are generally not authorised to use any medications to manage childbirth or PPH.[21] In the absence of national policy allowing these providers to use injectable oxytocin, access to gold standard measures for the prevention of PPH is limited.[21] Furthermore, the ability of MWs and AMWs to maintain a consistent cold chain for oxytocin is severely limited as most rural health centres or subcentres are not equipped with cold storage facilities.[23] Thus, out-of-facility deliveries attended by MWs or AMWs represent a setting where IOT could be beneficial through expanding access to first-line therapy and obviating the need for cold storage.

To guide product development and implementation planning activities, this study assessed the operational feasibility and acceptability of IOT for improving community-based care in Myanmar.

## METHODS

A qualitative inquiry using focus group discussions and in-depth interviews was conducted to gain insight into the perspectives of healthcare providers, community members and key informants. Methods are reported according to the Consolidated Criteria for Reporting Qualitative Research checklist.[24]

### Study setting and participants

Three administrative areas in Myanmar, South Dagon, Ngape and Thanlyin townships, were purposively selected for the study, based on their geographic location (rural/urban) and extent of use of the AMW cadre, and after discussion with Ministry of Health officials. South Dagon is an urban township and has a population of 297 000 with

one 50-bed township hospital, one station hospital and two urban health centres. Thanlyin is a large township, which has both urban and rural characteristics and has a population of 188 000 with a 150-bed district hospital, one station hospital and five rural health centres. Ngape is a rural township and has a population of 46 500 and has one 25-bed township hospital, one station hospital, two rural health centres and 14 subcentres. South Dagon and Thanlyin are located in Yangon region, where an average of 94.6% of women engage a skilled provider (including an MW) for antenatal care and the average number of visits is approximately four.[22 25] While only 65.4% of women give birth in a health facility, coverage of skilled attendance at birth is 82.5%.[22] Ngape is situated in Magway region, where 82.5% of women engage a skilled provider for antenatal care and the average number of visits is four.[22 25] Only 37.5% give birth in a health facility, however, 68.4% engage a skilled provider and a further 6.8% of deliveries are attended at home by an AMW.[22]

A total of 16 in-depth interviews and 11 focus group discussions were conducted involving 100 persons. Focus groups had between 5 and 10 participants and included a total of 26 MWs, 28 AMWs and 30 mothers with a child aged less than 3 years. In-depth interviews consisted of healthcare providers at the township level, obstetricians from both the public and private healthcare system and other key informants from the pharmaceutical industry and agencies working on maternal and child health. Key informants were selected on the basis of their knowledge and experience with the local health system and thus could provide insights into the considerations for new product introduction.

### Research team and data collection

The research team was led by an experienced qualitative Myanmar researcher. All six team members (three were medical doctors and the others had worked in the medical field) had previous experience with qualitative data collection. Apart from one, all were women. A 4-day training was given to all researchers on background, objectives and design of the study, and data collection, management and analysis methods. None of the research team members had a prior relationship with study participants. Rapport with the participants was built during focus group discussion through self-introduction and explaining the research objectives.

Data collection for in-depth interviews was mostly conducted in the participants' office as the place of choice by the interviewee ensuring confidentiality and convenience. Focus group discussions with MWs, AMWs and mothers were conducted in comfortable private places where confidentiality and privacy were ensured. Separate FGDs were conducted for different categories of participants. Question guides were used to investigate the practices around childbirth, beliefs around PPH and bleeding, perspectives of medications around the time of birth, oxytocin current practices, perspective towards IOT and possible implementation strategies. In all interviews and focus groups, only after the discussion moved towards management of PPH, the moderator gave a brief description of IOT. To aid in the description of the product, a Rotahaler device was shown as an example of a low-cost, passive inhaler, which could be used for inhalable oxytocin.

The duration of the focus groups and interviews ranged from 30 min to 90 min, and most were conducted in Myanmar language. Data were collected until it was felt that data saturation was obtained. All the focus group discussions were audio-recorded using a digital recorder. For the in-depth interviews, apart from three obstetricians and two key informants from the pharmaceutical industry who declined to be digitally recorded, all others were digitally recorded. Contemporaneous notes were taken for all discussions and interviews.

### Data management and analysis

All Myanmar language interviews and focus group discussions were transcribed in Myanmar language by the note taker for the focus group discussions and by the interviewer for the in-depth interviews and were checked against the notes for consistency and validity. Transcripts were translated into English by a translator and checked against the Myanmar transcripts to ensure the quality and the meaning of the original transcripts was not lost. To aid the analysis, NVivo (V.11, VMWare) software was used. Three data analysts coded the transcripts. Reliability coding was set at 80% agreement, and the inter-coder reliability was found to be over 80%. This approach balanced the differing views of the researchers in the study. Participant characteristic codes (obstetricians, hospital staff, MWs, AMWs and mothers) and setting codes (urban and rural) were also considered during the coding process. Before finalising the code structure, all the researchers who coded the transcripts collaboratively reviewed and agreed on the final structure. Qualitative content analysis using both inductive and deductive approaches was used in interpreting the findings.

### Patient and public involvement

This study sought to understand the perceptions of community members, healthcare providers and key informants towards the conceptual acceptability and feasibility of an IOT product for the prevention of PPH in community settings of Myanmar. No intervention was tested as part of this research, and the participants are not formally considered 'patients' in that they are not seeking or receiving medical care as part of this study.

However, the research described here is in itself a strategy to increase patient and public involvement in research. Through the conduct of this study, the product development team has created a mechanism by which the design and future implementation of the IOT product can be informed by the priorities, experiences and preferences of the target population for the product (ie, mothers and healthcare providers).

While the design, recruitment and conduct of this study was not directly informed by members of the public who represent the participant groups involved, findings from the study were disseminated back to the public through the conduct of a half day workshop in March 2017, which was attended by healthcare providers, government officials and members of the pharmaceutical industry.

## Ethics

Written informed consent was obtained from all participants, all of whom were reimbursed for the actual cost of travel plus a daily allowance to cover meal costs (3000 kyats, equivalent to US$2.5).

## FINDINGS
### Current perceptions and practices towards PPH in community settings
#### Cultural beliefs towards postpartum bleeding

Regardless of the place of residence, many women in the study preferred home births over facility births, mainly because of cost, the supportive family environment and inconvenience of engaging a facility. Women in the study described postpartum blood as 'bad' or 'impure' and explained that it was essential to remove this blood from the body in order to avoid subsequent pain or hardship. This viewpoint seemed more common among participants from rural areas as compared with their urban counterparts. A traditional oral medicine named 'Memakin say' was commonly used in the first 3 days after delivery to facilitate bleeding and clean the body.

> You see, we have to carry the pregnancy for 9 to 10 months and the bad blood accumulated in our body for so long. So with the birth of the baby, it should all come out… or else if it stays in our body, we will not be healthy. So bleeding is a necessity for us after delivery. If it does not come out…. pain in supra-pubic area will occur. (33-year-old mother of three, rural area)

Despite traditional beliefs surrounding postpartum bleeding, a prominent theme during discussions with mothers was the trust placed in the formal health sector for matters relating to maternal health. In general, rural participants saw community-based providers, such as MWs and AMWs, as the key decision makers or advisers for health, whereas participants from urban areas seemed to place more reliance on staff at healthcare facilities. For the majority of women in the study, MWs and AMWs were the main home birth attendants with occasional births attended by traditional birth attendants.

#### Burden of PPH, its management and associated challenges

Alongside the perception of the need to expel bad blood, excessive blood loss after childbirth was also cited as a danger by many community members and two mothers participating in the study had personally experienced excessive bleeding postpartum.

> We are afraid sometimes that after the birth of the baby, bleeding might occur. (33-year-old mother of four, rural area)

Mothers who had seen postpartum bleeding explained that dangerous levels of bleeding were recognised by monitoring how quickly a sarong (a traditional wrapped skirt for women) became soaked with blood. While many healthcare providers suggested that PPH is rare, others spoke of the serious nature of the condition and described excessive bleeding like a 'pipe water falling from a pipeline'. AMWs often described the challenges associated with PPH management in the context of their restricted authority to administer medications and their reliance on MWs who were often not readily accessible.

> I know bleeding is serious and the mother can die, but what shall we do apart from waiting. It was so long waiting for her (the MW) to arrive. I dare not to give injection. I am afraid of someone seeing me giving an injection to the patient. (AMW, rural area)

Another challenge faced by community-based home birth care providers is the law restricted towards injection. Healthcare providers explained that the current narcotic law in Myanmar restricts use of injections by MWs and AMWs such that they are prohibited from routinely delivering an oxytocin injection for the prevention of PPH when attending out-of-facility deliveries. However, in practice, some MWs are administering an injection of oxytocin with authorisation from the residing township medical officers due to the live-saving potential of the drug. In practice, the absence of national law authorising both MWs and AMWs to deliver injectable oxytocin restricts widespread use of oxytocin for PPH prevention.

> Another problem is that in our country, the narcotics laws do not allow the midwives to give (medicines in) injection form. So if we can avoid the injection form, and then we can use it (oxytocin)…and then it is definitely useful for prevention of PPH. (Obstetrician in private practice, urban area)

#### Access to quality oxytocin in community settings

Prevention and management of PPH was mainly performed by MW at the community level in which injection oxytocin was used. However, reliability and availability of the cold storage required to maintain oxytocin quality within the settings of community-based care was limited. Refrigerators were not available at rural health centres and subcentres, which are designed as the bases for community-based MWs. Community-based providers can keep items cold for short periods using ice bricks (supplied from the nearest township hospital) in vaccine transport carriers, but they explained the difficulty they would face if they were to maintain consistent refrigeration for oxytocin.

> We have to buy the ice in the city and there must be electricity to keep it cool. In the rainy season, it

[keeping oxytocin cold] would not be feasible. (MW, rural area)

In settings where cold storage facilities are available, unreliability of electricity supply was cited as a barrier, with frequent power outages and voltage fluctuation. Since most of the injection oxytocin were not kept in a cold storage, which may have effect on the quality.

## Acceptability and feasibility of IOT
### Conceptual acceptability
Acceptability was explored through the perceived need for and willingness to use the IOT product when available. Although general acceptance of all forms of medication prescribed by healthcare providers was high among the community, injections stood out to be most preferred due to their perceived rapid action. Positive views towards inhalers were also expressed by community members, who cited ease of use and a believed rapid onset of action as benefits. One woman saw the inhalant as a less painful alternative to injections.

We don't need to drink it and it does not hurt like an injection, I think the inhaler will be really good. (23-year-old mother, urban area)

A key informant from the pharmaceutical industry expressed concerns about the acceptability of the IOT product to community members, as past introduction of a dry powder inhaler for asthma had failed to result in sustainable uptake in Myanmar. This informant speculated that this was due to patients' concern about inhaling powder into the lungs. However, this concern was not heard from community members participating in this study and most mothers expressed their willingness to accept IOT compared with oral and injectable drugs. Some concerns were expressed by a minority of these stakeholders, most commonly, they worried whether they would know the correct time at which to use the drug, fearing that they would inhale too early (before the baby is delivered).

Mothers and healthcare providers alike expressed a strong desire for a single-use device packaged within a sealed sachet to keep it 'sterile' and to prevent transmission of infections. Although there was no preference to the colour, the smell of the drug was thought to have a significant impact on acceptability. Several MWs and AMWs and one obstetrician suggested that mothers would be reluctant to inhale the drug if it has a strong or unpleasant odour. However, there were some mothers who explained that while a 'soft' smell would be favoured, they would be willing and able to inhale the IOT product regardless, if it was important for their health.

Some MWs and AMWs held concerns about the feasibility of a mother inhaling from a device immediately after delivery, fearing the woman may be too tired. They suggested that only one inhalation or potentially a few with not too much force would be feasible for postpartum women. Other healthcare providers disputed this idea

and suggested that mothers would be completely able to use the inhaler as shown. This viewpoint was supported by mothers themselves, many of whom declared that they would have no problem using the device after delivery. Following a demonstration of the Rotahaler device during the interviews and focus groups, both community members and healthcare providers remarked on the ease of use of the inhaler.

This ease of administration was highlighted as a benefit of the IOT product for MWs and AMWs attending home births, who are often alone and struggle to manage all aspects of maternal and neonatal care that are required during the early postpartum period.

Actually, that (IOT) is more comfortable. It also reduces work. The injection to the butt will be like: open it, put it and such. This will be like open and can just say to the patient to inhale it by herself. (AMW, urban area)

Healthcare providers, especially the obstetricians, emphasised that the product should first be demonstrated to be effective with few side effects before it would be accepted by either themselves or the community. They suggested that assurance of safety and efficacy could come from WHO recommendations or from proven clinical trials before introduction of the product into routine clinical care.

Although I think IOT is useful, the best thing is to wait for the WHO package. They will have some trials and then they will find out the potency and efficacy and then safety and then after that they will recommend to use in developing countries. (Obstetrician in private practice, urban area)

### Administration of IOT
MWs and AMWs attending home births were suggested as suitable providers for the introduction of IOT. MWs were seen as possessing the skills and knowledge required for them to safely and effectively administer the drug or instruct the mother or a relative to do so. The ease of use of the IOT product was also thought to offer the potential to relieve the burden on MWs as they are often alone when attending deliveries in the community.

MW 4: 'We can do our management better [with IOT] during birth'.
MW 9: ' … [W]e can ask someone to help us give the inhalation to the mother, but we cannot do this with an injection as it needs a healthcare provider with skill to do it'. (FGD with MWs, urban area)

Opinions towards task-shifting IOT administration to AMWs were mixed among healthcare providers. In hard-to-reach areas, AMWs, who often reside in the remote villages, were suggested as a more accessible provider of essential care in the absence of an MW. Some MWs and obstetricians were in support of IOT use by AMWs and

suggested that, particularly in remote areas, it would be necessary to allow AMWs direct access to the product.

> I think it should be in the hands of AMW, especially in the hilly regions. In emergency situations mothers don't have anyone to rely on. (MW, rural area)

MWs themselves explained that they often attend deliveries in the absence of an MW, and in this situation, a non-injectable medication, such as IOT, would provide them the opportunity to protect the mother from a potentially fatal PPH.

Conversely, some of the township-level providers and MWs expressed their concern about the possibility of AMWs misusing IOT during delivery to augment labour, a practice suggested to be common among AMWs in the past. However, there were also suggestions that release of IOT with clear instructions and rebranding as a product indicated for postpartum use only (unlike the injection), could benefit women and community-based providers. One AMW suggested that the perception of oxytocin as a drug for augmenting labour is associated specifically with the injection and so may not be automatically applied to the inhaler.

> Moderator: 'Do people consider it [IOT] as a uterus opening drug?'.
> AMW: 'They think injection will open the uterus, not an inhaler, so I think it is ok'. (FGD with AMWs, rural area)

Regarding community distribution of IOT to mothers, the responses varied widely between the community and the healthcare providers. Mothers supported the idea, explaining that it would ensure they have the product regardless of where they deliver and who attends them.

> It would be more convenient if you can give it to the mothers, then we won't need to worry about getting the drug in time. Even if the Sayarma (MW) did not arrive in time, we can still use it after the delivery of the baby. It would be really useful. (28-year-old mother of three, rural area)

In contrast, healthcare providers were more opposed to community distribution of IOT due to the fact that mothers may use the product before labour or delivery. Some mothers and healthcare providers suggested that community distribution of IOT should be selective and targeted specifically to mothers in hard-to-reach areas, as they are most likely to be at risk of delivering outside a facility or in the absence of a skilled provider. In general, a prominent opinion among community members and healthcare providers was that particular care was needed for community distribution of IOT.

### Training

While there was some awareness of orally inhaled medicines among community members, none of the mothers participating in this study had personally used an oral inhaler. As such, all stakeholders stressed the importance of training mothers how to use the inhaler. All participants suggested that training for mothers on IOT should start during the antenatal period. They emphasised that early exposure to the device would be beneficial for mothers to understand its function and intended use. The local MW of each village was suggested as the most suitable person for training the mother.

For the training of healthcare providers, suggestions centred on the necessity to conduct systematic training to ensure all cadres of healthcare providers are adequately educated about the IOT product. In the current model, healthcare workers are trained through a cascade system, beginning with a training of trainers workshop at the national level and thereafter through states/regions, districts, townships and finally down to the community-level health workers. However, some MWs complained that this mechanism of training can result in messages being inaccurately or ineffectively passed down through the chain. Several MWs suggested that direct training where they can interact with teachers at the central level would be preferable to them. Additionally, a trainer at national level explained that the feasibility of conducting coordinated nationwide training would depend on finances and supplies. The suggested training content included indication for use, contraindications, side effects, possibility of repeated dose and storage requirements.

Clear instructions on the purpose of the drug, how to use the device and possible side effects either written in Myanmar language or pictorially depicted were suggested by some mothers and healthcare providers.

### IOT pricing

Reports from mothers and healthcare providers participating in this study suggested that in recent years, drugs and services are provided free of charge for women delivering in healthcare facilities. For out-of-facility deliveries attended by MWs, many of the respondents in the study mentioned that women pay for delivery with cash or gifts but that drugs such as oxytocin are not purchased directly by patients. This suggests that patients' out-of-pocket expenditure for oxytocin is currently minimal, and therefore, IOT may likewise be provided free of charge to end-users. However, some healthcare providers expressed concerns over the sustainability of current levels of government health financing, and thus, the possibility that user charges will be applied to IOT should be considered. As such, participants were asked to suggest a price for IOT to ensure its affordability in the event that patients are required to directly purchase the product. Most stakeholders felt that a price between 500 and 1000 kyats (US$0.40–US$0.80) would be most affordable to community members and that not more than 500 kyats (US$0.40) would be necessary to also ensure affordability to low-income families.

### DISCUSSION

To understand the acceptability and feasibility of implementing IOT in Myanmar, a situation assessment was

conducted to explore current attitudes and practices towards childbirth, PPH and oxytocin. IOT acceptability was explored from the point of view of community members and healthcare providers, while perspectives of all stakeholders were used to understand the operational feasibility. Responses to the IOT product were overwhelmingly positive from community members, all of whom expressed an unconditional acceptance of the product and praised its ease of use. These expressions of acceptance may in part be due to an appreciation for a product that replaces an injection, as the fear of pain associated with injectable products was a concern for some community members. Additionally, community members who were familiar with inhalers generally held positive views towards this route of administration, due to the rapid relief that is experienced after use of these medications for respiratory conditions (such as asthma). However, there may also be an element of social desirability bias in participant responses, and acceptability of the product to community members should be critically evaluated. For example, a barrier to the acceptability of IOT may arise from existing traditional beliefs and practices, which encourage expulsion of blood after childbirth. Community members may therefore resist a product that controls or reduces postpartum blood loss. However, many mothers from both urban and rural areas expressed concerns about excessive bleeding after childbirth, attitudes that could be harnessed to facilitate acceptance of IOT as a product for preventing dangerous levels of bleeding. Furthermore, community members predominantly place their trust in the healthcare system, particularly in MWs. Thus, it is likely that advice from MWs and other healthcare providers will dispel potentially inhibiting attitudes and facilitate community acceptance of IOT. These findings will help inform the design of education strategies in order to improve acceptability to beneficiaries. Specifically, for these settings, it will be important to inform community members about IOT in a way that is sensitive to existing beliefs, harnessing and encouraging enabling attitudes (such as recognition of the danger of excessive blood loss) and mitigating the impact of potential barriers (such as the traditional belief for the need of postpartum blood loss). Identification of the MW as a key advisor for health highlights that these providers would likely be an effective channel through which this information could be delivered to the community.

Among healthcare providers, opinions about IOT were predominantly positive, particularly in relation to ease of use. However, remarks from healthcare providers were often tempered with expressions of concern or apprehension. Given that IOT uses a different administration route, many providers expressed their desire to be assured of its efficacy and safety. Robust evidence of safety and efficacy will be generated as part of the regulatory requirements for the product. Attitudes of healthcare providers expressed in this study highlight the importance of making this data readily available to promote acceptance.

In addition to the safety and efficacy of the product, a common concern among healthcare providers was whether the device could be operated correctly to deliver the required dose to the patient. In this regard, care will need to be taken to ensure that community members can be appropriately trained on how to use the device given that the utility of the IOT product may to some degree be dependent on their participation in drug administration.

Given that home births remain common in Myanmar, we explored the acceptability of IOT use in these settings, with particular consideration for the possibility of product use by MW or AMW, who are the common attendants of out-of-facility deliveries. Most stakeholders suggested that IOT would be both appropriate and beneficial for use by MWs at out-of-facility deliveries. Healthcare providers suggested that MWs possessed the skills and knowledge required to administer the drug or instruct the mother or relative on administration. At a policy level, a non-injectable form of oxytocin aligns with both national drug law (which prohibits MWs use of injections) and standard treatment guidelines (which prioritise use of oxytocin over misoprostol for prevention of PPH). Attitudes towards the acceptability of product use by AMW were more varied. Although current policy states that AMWs are only to assist MWs, staff shortages and extreme geographic barriers severely limit accessibility of MWs, and as a result, AMWs are often required to act as the sole attendant for out-of-facility deliveries in rural remote areas.[21 26] However, national policy currently prohibits use of uterotonics by AMWs, and many stakeholders participating in this study had reservations about tasking-shifting IOT to these providers.[27] Objections were primarily based on fears that the drug would be misused for the induction or augmentation of labour, which was reported to be a common practice among AMWs in the past. In contrast, there were some stakeholders, including township-level policymakers, who suggested that the knowledge and skill set of AMWs is improving over time and that these providers should be considered for IOT administration. Training for end-users can be considered in the context of training healthcare providers and community members in the use of IOT. The coordinated chain of training down the hierarchy of healthcare providers currently in place in Myanmar was described as a means to efficiently and effectively ensure all healthcare providers in the country receive appropriate training. However, assuring the consistency of these trainings may be a challenge and may require both financial and system support.

Training pregnant women and their families through antenatal care appears to be highly feasible as coverage of at least one antenatal visit among women in Myanmar is high (81%).[22] However, given the expressed unfamiliarity of community members towards inhalers, repeated trainings may be necessary. Compared with one antenatal visit, attendance of at least four sessions is less common, particularly in rural areas where 51% of women attend at least four visits, compared with 84% of their urban counterparts.[22] Thus, mechanisms to ensure adequate

comprehension of device operation among community members, for example, through the use of hands-on training exercises, will be a critical aspect of product introduction strategies. Additionally, this study suggests that there is a strong degree of community trust in and compliance with the advice of healthcare providers, further highlighting the availability of effective channels for community education.

## CONCLUSION

This study has uncovered a variety of factors that can be considered for the operational feasibility and acceptability of IOT product in community-based care settings in Myanmar. Use of IOT at community births attended by a MW was acceptable to most stakeholders, and the ease of product use, heat stability and non-injectable delivery route were considered to be the major facilitators. Further decentralisation of services to AMWs could further increase uptake of IOT, particularly in remote rural areas. The trust community members place in healthcare personnel, particularly community-based providers, may help facilitate community acceptance of IOT. Ensuring comprehensive training to community-based providers and mothers through participatory approaches starting from the antenatal period will also enhance the operational feasibility of the IOT product into the community.

**Author affiliations**
[1]Burnet Institute, Melbourne, Australia
[2]Department of Medicine, Royal Melbourne Hospital, University of Melbourne, Melbourne, Australia
[3]Monash Institute of Pharmaceutical Sciences, Monash University, Melbourne, Australia
[4]Department of Epidemiology and Preventive Medicine, Monash University, Melbourne, Australia
[5]International Centre for Reproductive Health, Department of Obstetrics and Gynaecology, Ghent University, Ghent, Belgium

**Acknowledgements** The authors would like to acknowledge the Department of Public Health, Ministry of Health and Sports, Myanmar, for the collaboration and all the administrative support given by all the health staff in the three study townships. We are also grateful to Burnet Institute Myanmar research team members (Kyaw Soe Thant, Tin Tin Wai and Thandar Aye) for kindly assisting in data collection. We would also like to thank all the stakeholders who participated in the study without whom this study would not have been possible.

**Contributors** KKT contributed to study design, data collection, data analysis and led the first draft and finalisation of the manuscript. VO contributed to study design, data collection, data analysis and development of the manuscript. YM contributed to study design and data analysis and development of the manuscript. TL contributed to data collection and development of the manuscript. MM and PL contributed to study design and development of the manuscript. SL contributed to study design, data analysis and led the revisions of the manuscript. All authors read and approved the final manuscript.

**Funding** This study and report was made possible by the generous support of the Saving Lives at Birth partners: the United States Agency for International Development (USAID), the Government of Norway, the Bill and Melinda Gates Foundation, Grand Challenges Canada and the UK Government (grant number: AID-OAA-F-14-00046).

**Disclaimer** GlaxoSmithKline had no role in the funding, design or conduct of this study. The authors have no commercial interest in the outcomes of this study or the introduction of the inhaled oxytocin product in low and lower middle-income countries.

**Competing interests** We have read and understood BMJ policy on declaration of interests. VO, PL and MM are part of a product development team at Monash University, which is progressing the development of a heat stable oxytocin product for the prevention of PPH in resource-poor settings. MM is the coinventor of a worldwide patent application 'Method and Formulation for Inhalation' (WO 2013/016754) that covers the delivery of biologically active agents (including oxytocin) in the form of dry powders for inhalation. Inhaled oxytocin is being developed through a product development collaboration agreement between Monash University and GlaxoSmithKline. Authors declare no other conflict of interest.

**Patient consent** Not required.

**Ethics approval** Ethical clearance for the study was obtained from the Alfred Hospital Ethics Committee, Australia (Project 153/15), the Monash University Human Research Ethics Committee, Australia (CF15/1701 – 2015000854) and from the Ethical Review Committee on Medical Research Involving Human Subjects from the Department of Medical Research Myanmar (48/ Ethics 2015).

**Provenance and peer review** Not commissioned; externally peer reviewed.

**Data sharing statement** The data generated and analysed for the current study are not publicly available as the qualitative study was collected from specific townships, and the information may be identifiable to particular individuals, risking a breach in confidentiality.

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
