## [Reviewer comments · BMJ Open]

ARTICLE DETAILS

TITLE (PROVISIONAL)	Assessing the operational feasibility and acceptability of an inhalable formulation of oxytocin for improving community-based prevention of post-partum haemorrhage in Myanmar: A qualitative inquiry
AUTHORS	Than, Kyu Kyu; Oliver, Victoria; Mohamed, Yasmin; La, Thazin; Lambert, Pete; McIntosh, Michelle; Luchters, Stanley

VERSION 1 – REVIEW

REVIEWER	Narjis Rizvi Department of Community Health Sciences Ada Khan University Karachi Pakistan
REVIEW RETURNED	01-Mar-2018

GENERAL COMMENTS	Overall This manuscript is more like a product marketing/promotion report. The manuscript is silent about the efficacy and safety of the product that it intends to promote through an assessment of feasibility and acceptability. In order to assess the feasibility and acceptability the manuscript has to provide evidence of its efficacy and safety as pre-requisite. Title The phrase in the title “Improving community-based care in Myanmar” –is non-specific and does not reflect the specific study objective “prevention of post-partum hemorrhage”. Abstract 1. Concise, coherent and well-structured2. Page 2/17, the sentence in line 25 in the result section in the abstract “Community-based providers such as midwives and volunteer auxiliary midwives were described as key advocates for promoting community acceptance of the product.”- It is not clear who were the respondents.3. Page 2/17, conclusions are missing under the conclusion section. Under the title of conclusions actually recommendations are stated. Strengths and Limitations 1. Page 3/17, bullet 2 stating “Relatively large and diverse study population”- It is not a strength as this was a qualitative study. In this qualitative study design, number of participants is not the
---

strength since saturation determines the numbers, rather the appropriate selection is a strength.

2. Page 3/17, bullet 3 stating “purposively selected based on urban and rural” is a strength rather than limitation

3. Page 3/17, similarly bullet 3 stating “findings might not be representative of the perspectives of equivalent stakeholders in other areas in Myanmar”. It is an inappropriate statement since generalizability is not an expectation from qualitative studies.

Background

1. Page 3/17, the 3rd paragraph is clearly “Marketing” of a product

2. The background should rather have a paragraph on the efficacy and safety of the product with scientific evidence

Methods

1. Page 4/17, the 1st paragraph line 13 says “key informants”. It would be good to describe who were the key informants. And why were these considered “Key Informants”

2. Page 4/17, under the title “Study Setting and Participants”-the 1st four lines 11-15 would better fit into the introduction.

3. Page 4/17, under the title “Study Setting and Participants”-line 41-what was the rationale to include personnel from pharmaceutical industry and agency as key informants?

4. Page 5/17, line 6 & 7 describe who were FGD participants. Please mention whether the FGD were conducted jointly or separate for different categories of participants.

5. Page 5/17, lines 15 & 16 are written as this was a group discussion to promote and market the product

Results

1. Page 7/17, there is a big jump from 1st-sub-theme “Home Births and Cultural Beliefs towards Post-Partum Bleeding” to the 2nd-Sub-theme “Access to Quality Oxytocin in Community Setting”

2. There should be some information with regard to PPH in relation to perceptions about the burden, management and challenges.

3. Page 9/17, quote stated in lines from 1-6. It is surprising to see that participants preferred inhalation over injections whereas literature reports that community members prefer injections as a mode for drug administration

4. Page 12/17, lines 1-6, two different prices for the product suggested by the participants are mentioned. Did participants reach to a consensus? If yes, what was the final suggestion? If no, which price is suggested by whom?

5. Results should also mention myths and misconceptions around PPH

	Discussion  1. Page 12/17, 1st paragraph lines 16-21 are in fact objectives. 2. Page 12/17, 2nd paragraph from 22-42 would fit better in introduction/background 3. Page 13/17, line 36 talks about task shifting. This was neither the objective of this research nor was in the scope of this work 4. In the discussion, scientific evidence either in support or against the study findings is missing all together. 5. In the discussion, no evidence is provided on efficacy, safety and effectiveness of the product Conclusion  1. The conclusion is mainly describing the participant's positive responses related to the feasibility of the introduction of the product 2. It is also giving steps for the implementation 6. Page 14/17, line 22-23 is mentioning "decentralization of services". This was neither the objective of this research nor was in the scope of this work
--	--

REVIEWER	Jo Durham University of Queensland Australia
REVIEW RETURNED	07-Mar-2018

GENERAL COMMENTS	This is an interesting paper but in places I feel it needs some reorganisation and additional background information. While it may be the first qualitative inquiry exploring operational feasibility of IOT, the novelty of the study and what it adds to product development is not that clear with most of the findings very similar to other findings on PPH prevention (i.e. beliefs about postpartum bleeding, trust in health workers, preference for those higher up in the systems, need for pictorial instructions, barriers in accessing facility-based birthing etc) Background The background discusses the challenges in low resource settings in administering oxytocin but could briefly present a discussion of oral administration of misoprostol which has been found to be effective, acceptable and able to be administered by lower level cadre of women themselves in the absence of a SBA, while it doesn't have to be at length this would help to contextualise the present study and the advantages of IOT. While IOT might simplify the administering oxytocin and the challenges of maintaining the cold chain etc can you briefly describe its advantages compared to oral administration of misoprostol? Methods Under background or study setting a brief description of the health system organisation would be helpful to understand where the MW and AMWs fit in. Also of payment – do women who deliver in facilities or are given oxytocin pay? How were the study sites purposively selected? Based on what criteria? How many women in the study sites receive the recommended
--

	number of ANC visits? How many women in the study sites deliver at home in the absence of a SBA? Findings The concept of “bad” blood is not unusual, how did women recognise excessive bleeding? What is Memakin Say? Is there likely to be any interaction between Memakin Say and IOT? How would the concept of bad blood be integrated into the messages about IOT to increase acceptability? (for discussion) If women in urban areas prefer the health facility does this have implication for targeting? Or are you proposing IOT for facilities in urban areas as well due to irregular electricity? I am not clear on this point TBAs are mentioned in passing on p. 8 can they administer IOT? Under conceptual acceptability – aside from the demonstration in the study were women familiar with the concept of inhaling medication? Are they familiar with inhalers? If they have had no experience would you expect them to express any disadvantages – especially to outsiders? Were none of the women curious about side-effects? Pg 11 is community distribution and advanced distribution the same or different? All participants said training should start during ANC – how many ANC visits is normal? What is the drop out rate? Wouldn't the 4th ANC visit be the most appropriate? How many women achieve 4 ANC visits in the study areas? As participants were asked about how much they would pay – do they normally pay for delivery with a MW and oxytocin? If people are willing to pay can it be distributed in advance via pharmacists? Discussion The second paragraph does not seem to add anything and can be integrated into the background or study setting Responses to IOT were overwhelmingly positive – there does not seem to be any critical reflection on this Like the healthcare providers if women and their families are not familiar with inhalers I also wonder if the community members would be able to use it correctly if they really were overwhelmingly positive or were being polite Overall, I feel the discussion can be more defined and specific – at the moment I am unclear what the really take away messages are or what you would really do in practice to inform product development and roll out – task shifting, training, pictorial instructions, trials and transparency in results etc are not particularly novel to any intervention - there can also be more recognition that what you found is not significantly different to what has been identified in regards to oral administration of misoprostol and that the barriers to facility based birthing are common and well documented Reflection on the limitations of the study and potential bias required
--	--

VERSION 1 – AUTHOR RESPONSE

Reviewer 1: Narjis Rizvi, Department of Community Health Science, Aga Khan University, Karachi, Pakistan

Comment: This manuscript is more like a product marketing/promotion report. The manuscript is silent about the efficacy and safety of the product that it intends to promote through an assessment of

feasibility and acceptability. In order to assess the feasibility and acceptability the manuscript has to provide evidence of its efficacy and safety as prerequisite.

Response: Thank you for the comment and we appreciate the input. However, the development of the inhaled oxytocin product is currently at an early clinical trials stage with only limited pharmacokinetic data from non-pregnant female volunteers. Initial data indicates that comparable systemic exposure of oxytocin, to that achieved following a 10 IU intramuscular injection, can be delivered through inhaled delivery. While these data indicate that an inhaled product has the potential to be similarly effective to the current injection, further clinical trials are required and will be conducted to characterise the product fully and determine utility in low resource settings. Given that there are many examples of promising global health technologies that have failed to achieve significant uptake and impact, despite possessing the technical attributes required to address an unmet need, we feel that providing in-depth information about the feasibility and acceptability is already of value and can inform future implementation and scale-up. We have explained this now in more detail in the background section.

Comment: Title: The phrase in the title “Improving community-based care in Myanmar” –is non-specific and does not reflect the specific study objective “prevention of post-partum haemorrhage”.

Response: This has now been addressed and the title has been rephrased to: “Assessing the operational feasibility and acceptability of an inhalable formulation of oxytocin for improving community-based prevention of post-partum haemorrhage in Myanmar: A qualitative inquiry”

Comment: Abstract: Concise, coherent and well-structured

Response: Thank you for the comment.

Comment: Abstract: Page 2/17, the sentence in line 25 in the result section in the abstract “Community-based providers such as midwives and volunteer auxiliary midwives were described as key advocates for promoting community acceptance of the product.”- It is not clear who were the respondents.

Response: The sentence has been rephrased clarifying the respondents who have provided this information. The sentence now reads: “Responses from healthcare providers and community members highlighted that community-based providers such as midwives and volunteer auxiliary midwives would be key advocates for promoting community acceptance of the product.”

Comment: Abstract: Page 2/17, conclusions are missing under the conclusion section. Under the title of conclusions actually recommendations are stated.

Response: Based on the available transcripts and after careful analysis of the information, we feel it is appropriate to conclude that in Myanmar, where home births and access to cold storage is limited, IOT may serve as an acceptable and feasible intervention. We therefore propose to keep our conclusion as earlier suggested.

Comment: Strengths and Limitations: Page 3/17, bullet 2 stating “Relatively large and diverse study population”- It is not a strength as this was a qualitative study. In this qualitative study design, number of participants is not the strength since saturation determines the numbers, rather the appropriate selection is a strength.

Response: Thank you for this input and we have now rephrased the sentence to highlight the importance of diverse perspectives without mentioning the number as a strength, and reads: “The participants involved health care workers and stakeholders from various levels of the health care system and the community, and from both urban and rural areas to provide an insight into a range of different perspectives, which has been triangulated to enrich the findings”.

Comment: Strengths and Limitations: Page 3/17, bullet 3 stating “purposely selected based on urban and rural” is a strength rather than limitation

Response: This has now been integrated in the second bullet and described as a strength. It now

reads: “The participants involved health care workers and stakeholders from various levels of the health care system and the community, and from both urban and rural areas to provide an insight into a range of different perspectives, which has been triangulated to enrich the findings.”

Comment: Strengths and Limitations: Page 3/17, similarly bullet 3 stating “findings might not be representative of the perspectives of equivalent stakeholders in other areas in Myanmar”. It is an inappropriate statement since generalizability is not an expectation from qualitative studies.

Response: Thank you for the comment. It has been rewritten as “Since the study was conducted in selected townships of Myanmar, the views and options towards the IOT product may differ in other places of the country.”

Comment: Background: Page 3/17, the 3rd paragraph is clearly “Marketing” of a product

Response: Thank you for the comment and the paragraph is amended and now articulates the rationale for the study within a broader global health perspective, describing the potential public health significance as also recommended by WHO.

The authors wish to clarify that we view this research not as a marketing study (traditionally conducted by pharmaceutical companies to maximise sales for commercial products) but as public health research aimed at maximising the positive impact of a novel maternal health technology. This work has been driven by researchers at Monash University and the Burnet Institute who recognise that, particularly for health technologies for low resource settings, an early understanding of the perspectives of stakeholders within target territories will result in the design of products that better align with the needs of the user and the priorities of policy-makers, thereby increasing the likelihood of widespread adoption and impact.

To clarify, GSK had no role in the funding, design or conduct of this study, nor does GSK or any authors of this study have any commercial interest in the outcomes of this study or the introduction of the inhaled oxytocin product in low and lower-middle income countries

Comment: The background should rather have a paragraph on the efficacy and safety of the product with scientific evidence

Response: We have added a paragraph detailing the current status of the IOT product, and the available evidence towards its efficacy. The background now includes the following elaboration: “The development of the inhaled oxytocin product is currently at an early clinical trials stage with pharmacokinetic data in non-pregnant female volunteers indicating that comparable systemic exposure of oxytocin to that achieved following a 10 IU intramuscular injection can be delivered through inhaled delivery. While these data indicate that an inhaled product has the potential to be similarly effective to the current injection, further clinical trials are required and will be conducted to characterise the product fully and determine utility in low resource settings.”

Comment: Methods: Page 4/17, the 1st paragraph line 13 says “key informants”. It would be good to describe who were the key informants. And why were these considered “Key Informants”

Response: A more detailed description of the key informants has been provided. They are considered as key informants because they are the main health care providers involved in decision-making around the use and provision oxytocin, including obstetricians and gynaecologists, township level health care implementers, and drug distribution agencies such UNFPA and pharmacology industry representatives. The revised sentence now reads: “In-depth interviews consisted of healthcare providers at the township level, obstetricians from both the public and private healthcare system and other key informants from the pharmaceutical industry and agencies working on maternal and child health, who were selected on the basis of their knowledge and experience with the local health system and thus could provide insights into the considerations for new product introduction.”

Comment: Methods: Page 4/17, under the title “Study Setting and Participants”-the 1st four lines 11-15 would better fit into the introduction.

Response: Thank you for the comment. As suggested, we have now moved the sentence to the background section.

Comment: Methods: Page 4/17, under the title “Study Setting and Participants”-line 41-what was the rationale to include personnel from pharmaceutical industry and agency as key informants?

Response: The main rationale for including personnel from the pharmaceutical industry was to better understand the country’s supply chain management system and to triangulate the processes described by the users (Obstetricians and Gynaecologist and other township level key informants).

Comment: Methods: Page 5/17, line 6 & 7 describe who were FGD participants. Please mention whether the FGD were conducted jointly or separate for different categories of participants.

Response: FGDs were separately conducted for different categories of the participants and this has been added to the methods section.

Comment: Methods: Page 5/17, lines 15 & 16 are written as this was a group discussion to promote and market the product

Response: We appreciate the comments, but removing this information could mislead the reader, and therefore provides relevant methodological information on how we assessed feasibility and acceptability in the absence of a final IOT product.

Comment: Results: Page 7/17, there is a big jump from 1st-sub-theme “Home Births and Cultural Beliefs towards Post-Partum Bleeding” to the 2nd-Sub-theme “Access to Quality Oxytocin in Community Setting”

Response: We appreciate the suggestion and have reviewed the flow of information in this part of the results section. We have revised the first sub-theme to focus only on cultural attitudes towards bleeding, and have therefore slightly amended the title of the section. The next section discusses the perceived burden of PPH and the challenges associated with its management. A more specific discussion of the challenges associated with access to quality oxytocin then follows on from this.

Comment: Results: There should be some information with regard to PPH in relation to perceptions about the burden, management and challenges.

Response: Thank you for the valuable suggestion, this is now discussed under in a section titled “Burden of PPH, its management and associated challenges.” This includes a discussion of our findings relating to how community members and health care providers view PPH or excessive bleeding. It also touches on current management in terms of seeking facility-based services in time of emergencies and introduces the challenges for providers to manage the condition

Comment: Results: Page 9/17, quote stated in lines from 1-6. It is surprising to see that participants preferred inhalation over injections whereas literature reports that community members prefer injections as a mode for drug administration

Response: We agree that the majority of participants communicated a preference for injections, based on the perceived rapid action. We describe this in the first paragraph of the “conceptual acceptability section” by stating: “Although general acceptance of all forms of medication prescribed by healthcare providers was high among the community, injections stood out to be most preferred due to their perceived rapid action.” However, some women feared the pain of the injection and one woman saw the inhalable product as a potential painless alternative, which is of interest. We have now clarified that this involved one opinion.

Comment: Results: Page 12/17, lines 1-6, two different prices for the product suggested by the participants are mentioned. Did participants reach to a consensus? If yes, what was the final suggestion? If no, which price is suggested by whom?

Response: Thank you for the comment. The price identified range from 500 to 1000 kyats. We have

rephrased the sentence to clarify that a price point of 500 kyats or below was suggested to ensure affordability for the poorer communities of the country.

Comment: Results should also mention myths and misconceptions around PPH

Response: Cultural attitudes towards bleeding (which in this context can be considered myths or misconception about PPH) have been discussed in the section on “Cultural beliefs towards postpartum bleeding”.

Comment: Discussion 1: Page 12/17, 1st paragraph lines 16-21 are in fact objectives.

Response: Thank you and indeed these first lines of the discussion summarise the objectives. In our opinion, we feel that reiterating the objectives of the study helps to set the context for the discussion section, and assists the reader. We propose to keep the current structure, but are willing to reconsider if the peer reviewer sees this as inappropriate.

Comment: Discussion 2. Page 12/17, 2nd paragraph from 22-42 would fit better in introduction/background

Response: Thank you for the comment. This paragraph has been revised and some of the information has been added to the background section of the manuscript, as suggested by the reviewer.

Comment: Discussion 3. Page 13/17, line 36 talks about task shifting. This was neither the objective of this research nor was in the scope of this work.

Response: We have included further detail to explain that as part of assessing acceptability, we considered the acceptability of product use for specific delivery settings (ie for home births) and thus explored acceptability of product use by healthcare providers who work in these settings (ie MW and AMW).

Comment: Discussion 4. In the discussion, scientific evidence either in support or against the study findings is missing all together.

Response: This is the first exploratory study regarding the novel inhaled oxytocin product for PPH prevention. As such, there is limited scientific evidence for comparison in terms of acceptability and feasibility of inhaled compound for this indication.

Comment: Discussion 5. In the discussion, no evidence is provided on efficacy, safety and effectiveness of the Product.

Response: Published studies investigating the efficacy, safety and effectiveness of the product have been introduced in the Background section. To clarify, the current study sought only stakeholder perspectives toward the concept of the product and thus a discussion of the safety and efficacy of the product was not a focus for this section of the manuscript.

Comment: Conclusion 1. The conclusion is mainly describing the participant’s positive responses related to the feasibility of the introduction of the product.

Response: As this addresses the main objective of the study regarding product acceptability and feasibility, these seem appropriate to conclude.

Comment: Conclusion 2. It is also giving steps for the implementation

Response: The authors aimed to have in-depth perspectives around acceptability and feasibility of this new compound to inform potential future implementation. Therefore, a description of the potential implication of our findings in terms of implementation seems appropriate.

Comment: Conclusion 3. Page 14/17, line 22-23 is mentioning “decentralization of services”. This was neither the objective of this research nor was in the scope of this work.

Response: The potential of decentralization of services (i.e. oxytocin use), particularly in the Myanmar

context where the majority of birth occur at home is essential to this study. As described, AMW are currently not authorized to provide oxytocin because it is currently delivered through injection. A future inhalable compound could potentially allow for decentralization of services from hospital-based oxytocin provision to also include community-based provision.

Reviewer: 2 (Jo Durham, University of Queensland, Australia)

Comment: This is an interesting paper but in places I feel it needs some reorganisation and additional background information. While it may be the first qualitative inquiry exploring operational feasibility of IOT, the novelty of the study and what it adds to product development is not that clear with most of the findings very similar to other findings on PPH prevention (i.e. beliefs about postpartum bleeding, trust in health workers, preference for those higher up in the systems, need for pictorial instructions, barriers in accessing facility-based birthing etc)

Response: Thank you for this input. This is indeed the first study to provide in-depth information on acceptability and operational feasibility of an inhalable oxytocin compound. We agree that the implications for product development might be limited, although some perspectives could be of use. However, the study does provide important perspectives to future implementation of IOT. Even though results may not come as a surprise, this remains extremely relevant information. Moreover, the fact that this information is available before product release is of interest and encouraging further product development.

We have expanded the background information, now including more information on the status of product development, and hope to have addressed the concerns of the reviewer.

Comment: The background discusses the challenges in low resource settings in administering oxytocin but could briefly present a discussion of oral administration of misoprostol which has been found to be effective, acceptable and able to be administered by lower level cadre of women themselves in the absence of a SBA, while it doesn't have to be at length this would help to contextualise the present study and the advantages of IOT. While IOT might simplify the administering oxytocin and the challenges of maintaining the cold chain etc can you briefly describe its advantages compared to oral administration of misoprostol?

Response: Thank you for the valuable suggestion. The background of the study has been updated with more information on the advantages and disadvantages of misoprostol. The third paragraph now reads: "An alternative to injectable oxytocin is oral Misoprostol. This is a heat stable drug, which has been recommended by WHO in settings where injectable oxytocin is limited. However, it is less effective with more frequent side effects compared to injectable oxytocin and thus the WHO recommends that availability of misoprostol should not detract from efforts to make oxytocin universally accessible. In addition, uptake of the misoprostol has faced significant challenges in some territories due to policy-maker concerns around abortion misuse and reluctance to task-shifting of uterotonics to community-based healthcare providers, who have been the primary target for many misoprostol implementation programs."

We have also included the follow sentence to directly compare the value offering of inhaled oxytocin with misoprostol: "Inhaled oxytocin (IOT) therefore has the potential to extend upon the value proposition of misoprostol by providing the gold standard uterotonic in a format which is suitable for use in settings where access to a reliable cold chain or skilled birth attendant is limited."

Comment: Under background or study setting a brief description of the health system organisation would be helpful to understand where the MW and AMWs fit in.

Response: A brief description of the health system organization of Myanmar has been added into the background section. The section reads: "The Myanmar health system is divided into central-, state/division- and township-level services. At the township level, health care services are provided through a township hospital, station hospital, rural health centres and sub-rural health centres. A

Township Medical Officer is the person responsible for administrative and clinical decision-making. Maternal and child health services at the sub-centre and community-level are provided by a midwife (MW). Due to shortages in midwives, a volunteer cadre of auxiliary midwives (AMWs) at the community-level support the MWs with provision of delivery and maternal and child health services.”

Comment: Also of payment – do women who deliver in facilities or are given oxytocin pay?

Response: We have included the following text to describe our findings relating to out-of-pocket expenditure for oxytocin: “Reports from mothers and healthcare providers participating in this study suggested that in recent years, drugs and services are provided free of charge for women delivering in healthcare facilities. For out-of-facility deliveries attended by MWs, many of the respondents in the study mentioned that women pay for delivery with cash or gifts, but that drugs such as oxytocin are not purchased directly by patients. This suggests that patient out-of-pocket expenditure for oxytocin is currently minimal.”

This has been included in the pricing sections as a means of explaining our rationale for exploring affordable pricing for community members.

Comment: How were the study sites purposively selected? Based on what criteria?

Response: Study sites were purposively selected based on their geographic location (rural/urban) and extent of use of the AMW cadre, and after discussion with Ministry of Health officials. This has been clarified in the methods section.

Comment: How many women in the study sites receive the recommended number of ANC visits?

Response: Further details have been provided in the methods section to outline the coverage of ANC in the regions where study townships are situated (Yangon and Magwe). The following paragraphs have been included: “South Dagon and Thanlyin are located in Yagon region, where an average of 94.6% women engage a skilled provider (including a MW) for antenatal care and the average number of visits is approximately four” and “Ngape is situated in Magway region, where 82.5% of women engage a skilled provider for antenatal care and the average number of visits is four.”

Comment: How many women in the study sites deliver at home in the absence of a SBA?

Response: Further details have been provided in the methods section to outline rates of institutional deliveries in the regions where study townships are situated and the coverage of skilled assistance at birth.

Comment: Findings, The concept of “bad” blood is not unusual, how did women recognise excessive bleeding?

Response: Women in the study mentioned that excessive bleeding was a rare event and recognize excessive bleeding as soaking the sarong (a traditional dress for women) wet within a short period, or bleeding continuously for more than an hour. We have added the following text to describe this viewpoint: “Mothers who had seen postpartum bleeding explained that dangerous levels of bleeding were recognised by monitoring how quickly a sarong (a traditional wrapped skirt for women) became soaked with blood.”

Comment: What is Memakin Say? Is there likely to be any interaction between Memakin Say and IOT?

Response: Memakin Say is a traditional oral medicine used for blood purification which facilitates bleeding. There is no formal evidence of interaction between Memakin Say and IOT. However, it is important to note that Memakin Say is normally taken after the 1st day of puerperium, and therefore not delivered at the same time as oxytocin. This is now detailed in the relevant section.

Comment: How would the concept of bad blood be integrated into the messages about IOT to

increase acceptability? (for discussion)

Response: Thank you for highlighting an important discussion point. While we are not in a position to directly outline potential education strategies, we have included a paragraph to further discuss how our findings could be used to inform these processes. The following text has been included: “These findings will help inform the design of education strategies in order to improve acceptability to beneficiaries. Specifically, for these settings it will be important to inform community members about IOT in a way which is sensitive to existing beliefs, harnessing and encouraging enabling attitudes (such as recognition of the danger of excessive blood loss), and mitigating the impact of potential barriers (such as the traditional belief in the need for postpartum blood loss). Identification of the MW as a key advisor for health, highlights that these providers would likely be an effective channel through which this information could be delivered to the community.”

Comment: If women in urban areas prefer the health facility does this have implication for targeting? Or are you proposing IOT for facilities in urban areas as well due to irregular electricity? I am not clear on this point.

Response: We are considering the utility and feasibility of the inhaled oxytocin product across a range of settings, including facility-based use (in both urban and rural settings). As you suggest, we see that the product will also serve benefit in urban settings due to the irregularity of the electricity in these areas. However, for the scope of this manuscript we focus on the acceptability and feasibility of the product for community-based care (eg births taking place outside of healthcare facilities). We feel a discussion of the applicability of the product for facilities in urban areas would therefore deviate from the main focus of the manuscript.

Comment: TBAs are mentioned in passing on p. 8 can they administer IOT?

Response: Births with TBAs are rare in study townships as TBAs have been replaced by AMWs, and therefore government policies do not support the use of TBAs for child birth (including PPH management).

Comment: Under conceptual acceptability – aside from the demonstration in the study were women familiar with the concept of inhaling medication? Are they familiar with inhalers? If they have had no experience would you expect them to express any disadvantages – especially to outsiders? Were none of the women curious about side-effects?

Response: Thank you for the valuable comment. Aside from the demonstration, most of the women in the study did not have personal experience with orally inhaled medicines. However, some women in the FGDs have seen inhalers for management of asthma. Under the conceptual acceptability section we have included the following text to describe community members’ perception of inhaled medications: “Positive views towards inhalers were also expressed by community members, who cited ease of use and a believed rapid onset of action as benefits. One woman saw the inhalant as a less painful alternative to injections....A key informant from the pharmaceutical industry expressed concerns about the acceptability of the IOT product to community members, as past introduction of a dry powder inhaler for asthma had failed to result in sustainable uptake in Myanmar. This informant speculated that this was due to patient concern about inhaling powder into the lungs.”

We have included further detail about the level personal experience with inhalers under the section discussing training in order to preface the importance of training community members.

Some of the women in the study expressed concerns for side effects and we have revised the manuscript with some of the women’s concern about side effects.

Comment: Pg 11 is community distribution and advanced distribution the same or different?

Response: It is the same; however advanced distribution in the community means distribution to mothers directly. We have now only used “community distribution” and specified where this applies to women directly.

Comment: All participants said training should start during ANC – how many ANC visits is normal? What is the dropout rate? Wouldn't the 4th ANC visit be the most appropriate? How many women achieve 4 ANC visits in the study areas?

Response: Four ANC visits is recommended, but in rural areas four ANC visit is below 60 percent. In the study townships an estimated 80 percent of women received at least one ANC visit. Further detail has been added to the discussion section to discuss the rates of ANC visits (1+ compared with 4+) and the associated implications for training. The section now reads: "Training pregnant women and their families through antenatal care appears to be highly feasible as coverage of at least one antenatal visit amongst women in Myanmar is high (81%).¹⁸ However, given the expressed unfamiliarity of community members towards inhalers, repeated trainings may be necessary. Compared to one antenatal visit, attendance of at least four sessions is less common, particularly in rural areas where 51% of women attend at least four visits, compared to 84% of their urban counterparts. Thus mechanisms to ensure adequate comprehension of device operation amongst community members, for example through the use of hands-on training exercises, will be a critical aspect of product introduction strategies."

Comment: As participants were asked about how much they would pay – do they normally pay for delivery with a MW and oxytocin? If people are willing to pay can it be distributed in advance via pharmacists?

Response: Many of the respondents in the study mentioned that they pay for delivery in cash and in kind, but not specific for oxytocin. There is no government pharmacist in many places of Myanmar. The MW is the main contact person at the village level for both delivery and distribution of drugs. We have added further detail in the Inhaled oxytocin pricing section to describe the degree to which patients currently have to pay for oxytocin and delivery services. The following text has been included: "Reports from mothers and healthcare providers participating in this study suggested that in recent years, delivery care drugs and services are provided free of charge in healthcare facilities. For out-of-facility deliveries attended by MWs, many of the respondents in the study mentioned that women pay for delivery with cash or gifts, but that drugs such as oxytocin are not purchased directly by patients. This suggests that patient out-of-pocket expenditure for oxytocin is currently minimal and therefore IOT may likewise be provided free of charge to end users. However, some healthcare providers expressed concerns over the sustainability of current levels of government health financing and thus the possibility that user charges will be applied to IOT should be considered. As such participants were also asked to suggest a price for IOT to ensure its affordability in the event that patients are required to directly purchase the product."

Comment: Discussion, The second paragraph does not seem to add anything and can be integrated into the background or study setting

Response: Thank you for the comment and it has been moved to the background of the study.

Comment: Responses to IOT were overwhelmingly positive – there does not seem to be any critical reflection on this. Like the healthcare providers if women and their families are not familiar with inhalers I also wonder if the community members would be able to use it correctly if they really were overwhelmingly positive or were being polite.

Response: Further detail has been added to interpret community members' positive responses to the product and critically analyse the truthfulness or reliability of this feedback. In the discussion it now reads: "Responses to the IOT product were overwhelmingly positive from community members, all of whom expressed an unconditional acceptance of the product and praised its ease of use. These expressions of acceptance may in part be due to an appreciation for a product that replaces an injection, as the fear of pain associated with injectable products was a concern for some community members. Additionally, community members who were familiar with inhalers generally held positive views towards this route of administration, due to the rapid relief that is experienced after use of these medications for respiratory conditions (such as asthma). However, there may also be an element of

social desirability bias in participant responses, and acceptability of the product to community members should be critically evaluated. For example, a barrier to the acceptability of IOT may arise from existing traditional beliefs and practices, which encourage expulsion of blood after childbirth.”

Comment: Overall, I feel the discussion can be more defined and specific – at the moment I am unclear what the really take away messages are or what you would really do in practice to inform product development and roll out – task shifting, training, pictorial instructions, trials and transparency in results etc are not particularly novel to any intervention - there can also be more recognition that what you found is not significantly different to what has been identified in regards to oral administration of misoprostol and that the barriers to facility based birthing are common and well documented.

Response: Thank you for the comment. The discussion has been amended and now more clearly stipulates the take-away messages, in particular we have outlined how findings will feed into future product development and implementation planning activities. For example we have discussed how understanding traditional beliefs around bleeding can inform education strategies to improved acceptability of the product to community members.

Comment: Reflection on the limitations of the study and potential bias required.

Response: We have added a sentence in this regard stipulating the potential for social desirability bias. Also, the strength and limitations section addresses a number of the limitations, and we have added one stating: “Operational feasibility and acceptability of the inhaled oxytocin product was assessed with a similar drug delivery system, but in the absence of the final product and in the context of limited experience with inhalable medications, which could have attributed to overwhelmingly positive perspectives.”

VERSION 2 – REVIEW

REVIEWER	Narjis Rizvi Community Health Sciences, Aga Khan University
REVIEW RETURNED	06-Aug-2018

GENERAL COMMENTS	Despite revision, the manuscript is still like a product marketing/promotion report. One paragraph has been added though about the efficacy of the product whose feasibility and acceptability is assessed through this research. However, reliable references or studies are pre-requisite to assess feasibility and acceptability. Title Title has been changed Abstract 1. Respondents are added. 2. Conclusions are still not capturing the key findings. Strengths and Limitations 1. Page 3/40, Bullet 2 is modified 2. Page 3/40, bullet 3 is modified Background 1. Page 3/40, a paragraph on the efficacy and safety of the product with scientific evidence is added though and the 3rd paragraph has now made the 4th paragraph (page 4/40) which is still clearly “Marketing” of a product Methods
--

	1. Page 6/40, the 1st paragraph line 22 says “key informants”- Explained 2. Page 6/40, under the title “Study Setting and Participants”-the 1st four lines 19-24 would better fit into the introduction. 3. Page 6/40, under the title “Study Setting and Participants”-line 54- what was the rationale to include personnel from pharmaceutical industry and agency as key informants? Not explained 4. Page 6/40, line 47 -52 describe who were FGD participants. Explained Results 1. Page 9/40, although the title of the theme has been changed and a sub-theme on management of PPH has been added, the transition from cultural practices with regard to PPH, its management and “Access to Quality Oxytocin in Community Setting” is not smooth 2. There should be some information with regard to PPH in relation to perceptions about the burden, management and challenges. 3. Page 9/40, quote stated in line 54. It is surprising to see that participants preferred inhalation over injections whereas literature reports that community members prefer injections as a mode for drug administration 4. Results should also mention myths and misconceptions around PPH Discussion 1. Page 13/40, 1st paragraph lines 16-21 are in fact objectives. 2. Discussion is very organized; does not clearly describe key findings, the similarities/controversies in the literature with regard to these key findings, the alternate explanation in case of controversial finding and the way forward. 3. In the discussion, scientific evidence either in support or against the study findings is missing all together. 4. In the discussion, no evidence is provided on efficacy, safety and effectiveness of the product Conclusion 1. The conclusion is mainly describing the participant’s positive responses related to the feasibility of the introduction of the product 2. It is also giving steps for the implementation
--	--

VERSION 2 – AUTHOR RESPONSE

Thank you for the positive review regarding the paper titled “Assessing the operational feasibility and acceptability of an inhalable formulation of oxytocin for improving community-based prevention of post-partum haemorrhage in Myanmar: A qualitative inquiry”

The detail responses to each of the comments are provided below:

Comment: Despite revision, the manuscript is still like a product marketing/promotion report. One paragraph has been added though about the efficacy of the product whose feasibility and acceptability is assessed through this research. However, reliable references or studies are pre-requisite to assess feasibility and acceptability.

Response: The authors in the manuscripts have explained the nature of the study clearly in the introduction of the manuscript stating that this type of study has never been done with new product development. Many of the products have been developed without taking into consideration of the true

utility and end target users who are often from developing countries. This is the uniqueness of this study in which target user voices are taken into consideration in the development of the inhaled oxytocin product. The authors sincerely believe that this is not a product marketing/promoting report. Since this is the first study, it also has limited literature to compare. However, the authors tried to put in relevant literature as much as possible in both introduction and discussion of the study.

Title

Comment: Title has been changed

Response: Thank you.

Abstract

Comment:

1. Respondents are added.
2. Conclusions are still not capturing the key findings.

Response: The conclusion section of the abstract has been reworded to clearly capture the key findings.

Comment: Strengths and Limitations

1. Page 3/40, Bullet 2 is modified
2. Page 3/40, bullet 3 is modified

Comment: Background

1. Page 3/40, a paragraph on the efficacy and safety of the product with scientific evidence is added though and the 3rd paragraph has now made the 4th paragraph (page 4/40) which is still clearly "Marketing" of a product.

Response: We have explained in the first paragraph of the response to comments.

Comment: Methods

1. Page 6/40, the 1st paragraph line 22 says "key informants"-Explained

Response: Thank you for the comment.

Comment:

2. Page 6/40, under the title "Study Setting and Participants"-the 1st four lines 19-24 would better fit into the introduction.

Response: Thank you for the comment, however, we think that explanation of the study setting fits better in this section.

Comment: 3. Page 6/40, under the title "Study Setting and Participants"-line 54-what was the rationale to include personnel from pharmaceutical industry and agency as key informants? Not explained

Response: In line with the explanation provided in the manuscript, the rationale for inclusion of the personnel from the pharmacological industry and agency centred on their knowledge and experience with the introduction of products into the health system in Myanmar, and thus the insights these informants could share regarding potential barriers and enablers was valuable to the aims of the study. For example, a key informant from the pharmaceutical industry was able share insights into the past acceptability of inhaled medications in Myanmar (as discussed in the "Conceptual acceptability" section of the manuscript)

Comment: 4. Page 6/40, line 47 -52 describe who were FGD participants. Explained

Response: Thank you for the comment.

Results

Comment: 1. Page 9/40, although the title of the theme has been changed and a sub-theme on management of PPH has been added, the transition from cultural practices with regard to PPH, its management and "Access to Quality Oxytocin in Community Setting" is not smooth.

Response: Thank you for the valuable suggestion. We have included a paragraph on management of PPH to smooth and connect the paragraph which states as follow; "Prevention and management of PPH was mainly performed by MW at the community level in which injection oxytocin was used. However reliability and availability of the cold storage required to maintain oxytocin quality within the setting of community based care was limited" The first paragraph in this section has also been moved into the section in Burden of PPH, its management and associated challenges.

Comment: 2. There should be some information with regard to PPH in relation to perceptions about the burden, management and challenges.

Response: This information has been added on to the topic Burden of PPH, its management and associated challenges.

Comment: 3. Page 9/40, quote stated in line 54. It is surprising to see that participants preferred inhalation over injections whereas literature reports that community members prefer injections as a mode for drug administration

Response: Please note, we have described that injections were commonly cited by community members as the most preferred route of drug administration. However, numerous positive views towards inhaled medication were also expressed by community members.

Comment: 4. Results should also mention myths and misconceptions around PPH

Response: Misconceptions concerning PPH has been mentioned in the section on cultural beliefs towards postpartum bleeding.

Discussion

Comment: 1. Page 13/40, 1st paragraph lines 16-21 are in fact objectives.

Response: As we explained in our previous response to the reviewer comments, we feel that reiterating the objectives of the study helps to set the context for the discussion section, and assists the reader.

Comment: 2. Discussion is very organized; does not clearly describe key findings, the similarities/controversies in the literature with regard to these key findings, the alternate explanation in case of controversial finding and the way forward.

3. In the discussion, scientific evidence either in support or against the study findings is missing all together.

Response: Thank you for the positive comment. Since this is the first study exploring the acceptability and feasibility of a new innovation IOT, it was not possible to compare the similarities/controversies in the literature with regard to the key findings. The way forward for the study has been mentioned in the

conclusion of the study line 38-50 on page 14

Comment: 4. In the discussion, no evidence is provided on efficacy, safety and effectiveness of the product.

Response: In the background section we have described the evidence that has been generated to date to show the safety and efficacy of the drug. Since the aims of this study were to assess the conceptual acceptability and feasibility of the product, a discussion of drug efficacy or safety was not considered relevant to the discussion section of this paper.

Conclusion

Comment: 1. The conclusion is mainly describing the participant's positive responses related to the feasibility of the introduction of the product
2. It is also giving steps for the implementation

Response: Thank you for the comment.